# Lifestyle of Families with Children Aged 4–8 Years before and during Lockdown Due to COVID-19 Pandemic in Poland

**DOI:** 10.3390/ijerph192215064

**Published:** 2022-11-16

**Authors:** Elżbieta Szczepańska, Barbara Janota

**Affiliations:** 1Department of Human Nutrition, School of Health Sciences in Bytom, Medical University of Silesia in Katowice, Jordana 19 Street, 41-808 Zabrze, Poland; 2Doctoral School, Medical University of Silesia in Katowice, Poniatowskiego 15 Street, 40-055 Katowice, Poland

**Keywords:** lifestyle, COVID-19, family, nutrition

## Abstract

The aim of the study was to evaluate the lifestyle changes of families with children aged 4–8 years during lockdown compared to the time before the COVID-19 pandemic. The study was conducted among 1098 parents during the first lockdown in Poland. An originally developed questionnaire was used as the research tool. The Wilcoxon test was used to determine the frequency of differences in the lifestyle of parents and children before the pandemic and during lockdown. Differences were found in the frequency of healthy habits in the periods under investigation, both in the lifestyle of parents and children. A moderately healthy lifestyle was predominant among families with children aged 4–8 years during lockdown. The families’ lifestyle significantly changed in relation to the time before the pandemic. There is a need for lifestyle education for families with children to prevent the development of bad habits during and after the pandemic and isolation time.

## 1. Introduction

The COVID-19 pandemic, which has been present since 2020, is having an impact on people’s lives through numerous restrictions, including periodic lockdowns that make people spend more time at home. The pandemic-related restrictions potentially lead to lifestyle changes, including changes of eating habits in families. During the COVID-19 pandemic in Poland, changes in the consumption of selected food products were observed, which may have an impact on the health of the polish population [1,2]. Global data also shows that physical activity levels have decreased and sleep quality has worsened due to the pandemic [3]. Those changes may have been related to changes in economy, caused by the limited availability of catering services and limited earning opportunities at a time of isolation of the society [2]. Negative lifestyle changes may put individuals at risk of a number of diseases such as obesity, diabetes and other metabolic disorders [4].

Lifestyle is defined as modifiable behaviors and choices that affect health and it includes nutrition, physical activity, social life, sleep. Based on the possibility of modification of these elements, lifestyle medicine was created, which has preventing impact on the civilization diseases development [5]. A number of aspects contribute to a lifestyle that promotes health and well-being, such as a good diet, for example [6]. Adults should consume 4–5 meals daily, while children should eat 5 meals with 3–4 h intervals between them [6]. This helps maintain normal blood glucose levels preventing episodes of sudden insulin secretion that could contribute to the development of type 2 diabetes in the future [4,7].

The need to eat fruit and vegetables in every meal by both children and adults stems from the fact that substances contained in fruit and vegetables are associated with numerous benefits [8]. Fruit and vegetables are a source of bioactive substances and prevent deficiencies in the body [9]. They are also a valued source of bioflavonoids, which have anti-oxidant effects. As a result, fruit and vegetables prevent the development of inflammation-related diseases. Fibre is an equally important ingredient of fruit and vegetables. An optimal dietary fibre intake decreases cardiovascular risk: fibre slows down cholesterol absorption in the digestive tract, thus reducing blood cholesterol level [10]. In addition, fibre has prebiotic effects and as such it promotes the growth of probiotic bacteria [11].

Dietary simple sugars, usually ingested in the form of sweets, sweet drinks and confectionery, contribute to the development of inflammation leading to chronic diseases. Simple sugar consumption should cover no more than 10% of the total daily energy demand [12]. The presence of sugar in children’s diet results in excessive energy intake, which leads to the development of overweight and obesity. These problems are particularly likely during lockdown, which causes physical activity of society to decrease, thus reducing the number of opportunities to burn kilocalories [13,14]. Excessive simple sugar consumption is a serious risk factor for dental caries, which is mainly associated with a high rate of sweetened drink consumption among children [15]. Water should be the source of dietary fluid. The latest recommendations indicate that adults need to drink 2000–2500 mL of water daily, while children aged 4–8 years should drink between 1250 mL and 1750 mL [16].

The consumption of fast food and savoury snacks provides dietary salt and saturated fatty acids, including trans fatty acids. Excessive salt intake leads to the development of arterial hypertension, while excessive consumption of saturated fats, particularly palmitic acid, decreases insulin sensitivity, potentially leading to insulin resistance and increasing the risk of atherosclerosis [17]. Saturated fatty acids activate cytokines such as IL-6, IL-18 and TNF-α, which are responsible for inflammatory processes [18].

Alcohol consumption has a negative impact on health. It causes gastrointestinal problems and compromises the immune system, potentially leading to the development of cancer, particularly pharyngeal, oral, laryngeal and colon cancer [19]. It is estimated that in 2016 in the European Union, 80,000 people died of alcohol-related cancer [19]. Moreover, alcohol consumed in the evening disrupts the length of different phases of sleep, resulting in chronic insomnia and compromised regeneration [18,20]. There are also known harmful effects of alcohol on the nervous and hormonal systems such as behavioral and emotional disorders [21].

Apart from healthy eating, it is equally important to sleep the recommended number of hours, perform physical activity, use electronic devices sensibly and take care of one’s emotional well-being. According to the US National Sleep Foundation recommendations, adults should sleep 7–9 h and children aged 4–8 years should sleep 10–13 h every day [22]. Inappropriate sleep duration contributes to increased cardiovascular risk, among other problems, and to compromised dietary habits, resulting in metabolic programming that is conducive to the development of diet-related diseases [23,24]. Moreover, it has been suggested that a regular rhythm of the day and food consumption will improve sleep and reduce insomnia [25].

An optimal number of minutes of physical activity supports normal cardiovascular function, carbohydrate metabolism and mental well-being among both children and adults [26,27]. WHO recommends that adults undertake 150–300 min of moderate-intensity or 70–150 min of vigorous-intensity physical activity per week [28]. According to WHO recommendations, children who are 5 years old or older should perform at least 60 min of physical activity of various intensity daily and undertake more intensive physical activity three times a week [28].

The restrictions associated with the pandemic made children and adults spend many hours in front of computer screens due to the need for remote education and work, and taking care of children receiving remote school instruction. Long hours of work in front of a computer screen and limited possibility to leave home and undertake physical activity undoubtedly elevate the risk of spinal diseases and short-sightedness, and may contribute to body weight increase [29,30,31].

Due to the COVID-19 pandemic and the associated lockdowns, it seems likely that the lifestyle of families with children has changed. The aim of the study was to evaluate the lifestyle of families with children aged 4–8 years during lockdown and to answer the question whether it has changed significantly compared to the time before the pandemic. It was assumed that the families’ lifestyles deteriorated during the quarantine.

## 2. Materials and Methods

The study was conducted among 1134 parents of children aged 4–8 years, in 2 first weeks of May 2020, at the lockdown associated with the COVID-19 pandemic in Poland. Only 1098 correctly completed questionnaires were analyzed. An originally developed questionnaire was used as the research tool, which was designed deductively. The questions were created in Polish language, in relation to the nutrition standards for the Polish population, Sleep Foundation recommendations and World Health Organization 2020 guidelines on physical activity and sedentary behavior [16,22,28]. The authors of the nutrition standards are the National Institute of Public Health—National Institute of Hygiene experts. The research tool was sent via the Internet using Facebook, where the survey was distributed among groups associating Polish parents of children in preschool and school age. In those groups, parents of children aged 4–8 were asked to fill in the questionnaire. Completing the survey was voluntary. It was assumed to collect the most possible number of filled questionnaires in the in the most possible number of groups. The collection was completed when no even one questionnaire had been collected throughout the week. The questionnaire contained a demographic section with questions regarding age, education, parents’ employment, and the number of children and their age, among other data. The research section included questions about the parents’ and children’s lifestyle during lockdown and the time before the pandemic. The data were collected and processed in Polish language, using Microsoft Excel 2010 and Statistica 13. There were also two steps taken to clean the data, by data validation using Microsoft Excel and double check by authors. Among children the following behaviors were considered as correct: eating 5 meals a day, making 3–4 h breaks between meals, eating vegetables several times a day and eating fruit at least once a day, eating: sweets, salty snacks, fast food products, sweet drinks occasionally or less frequently, using electronic devices up to 1 h a day, practicing physical activity for at least 1 h a day and sleeping 10–13 h a day. In the authors’ concept, the “occasionally” answer means less frequent than several times a month. Among parents the following behaviors were considered as correct: eating 4–5 meals a day, making 3–4 h breaks between meals, drinking at least 6 glasses of water, eating vegetables several times a day and eating fruit at least once a day, eating: sweets, salty snacks, fast food products, sweet drinks occasionally or less frequently, not consuming the alcohol, using electronic devices up to 2 h a day, practicing physical activity for at least 30 min a day and sleeping 7–9 h a day. The families’ lifestyle was assessed on the basis of the above guidelines, where each correct eating behavior was 1 point scored and the incorrect eating behavior was not scored. Then, the obtained points were summed up. The following percentage thresholds were adopted to assess the parents’ and children’s lifestyles: <40% of correct behaviour—an unhealthy lifestyle, 40–70% of correct behaviour—a moderately healthy lifestyle, >70% of correct behaviour—a healthy lifestyle. The differences between family lifestyles in the periods of time under investigation were determined using the Wilcoxon test. All results with a *p* < 0.05 were considered statistically significant.

## 3. Results

### 3.1. Characteristics of the Study Group

In the parents’ group, 34.5% were 20–30 years old, 58.0% were 31–40 years old and 7.5% were 41–50 years old. The group of children was consisted of children aged 4–8. Among the people who participated in the study, the largest group were parents with higher education (n = 614; 55.9%), not working (n = 529; 48.2%) and having 2 children (n = 572; 52.1%).

### 3.2. Meal-Related Behaviours

Selected meal-related behaviour is presented in Table 1.

Both before the pandemic and during lockdown, parents usually ate 4–5 meals (50.5% and 66.0%, respectively) and kept 3–4 h intervals between meals (57.9% and 55.7%, respectively). Before the pandemic and during lockdown, most children ate 5 meals daily (50.1% and 49.2%, respectively) and kept 3–4 h intervals between meals similar to their parents (63.5% and 59.4%, respectively).

Statistically significant differences were found in the frequency of meals and the duration of intervals between them in the two study groups between the periods under investigation (Table 1).

### 3.3. Frequency of Consumption of Selected Products

The frequency of consumption of selected products by parents and children is presented in Table 2 and Table 3.

Before the pandemic and during lockdown, vegetables were usually eaten a few times a day (28.5% and 30.2%, respectively), fruit were eaten once a day (30.6% and 28.6%, respectively) and sweets were mostly eaten occasionally (26.3% and 26.8%, respectively). Similar frequency was found for savoury snacks (37.5% and 30.1%, respectively) and fast-food products (60.7% and 45.7%, respectively). Before the pandemic, alcoholic beverages were usually consumed occasionally (46.2%), while during lockdown most individuals declared not consuming alcohol at all (42.6%).

Statistically significant differences were found in the frequency of vegetable, fast food and alcohol consumption by parents between the periods under investigation (Table 2).

Before the pandemic and during lockdown, most children ate vegetables once a day (27.1% and 27.2%, respectively). Before the pandemic, children usually ate fruit once a day, while during lockdown, children ate them a few times a day (32.9% and 32.2%, respectively). Before the pandemic and during lockdown, the majority of children consumed sweets a few times a week (26.7% and 26.9%, respectively) and savoury snacks occasionally (38.2% and 37.3%, respectively). Fast food products were usually eaten occasionally before the pandemic, while during lockdown, most children did not eat such products at all (48.3% and 45.5%, respectively). Both before the pandemic and during lockdown, the majority of children did not consume any sweetened drinks at all (55.1% and 55.9%, respectively).

Statistically significant differences were found in the frequency of fruit, sweet, savoury snack, fast food and sweetened drink consumption by children between the periods under investigation (Table 3).

### 3.4. Selected Aspects of Lifestyle

Selected aspects of parents’ and children’s lifestyles are presented in Table 4.

The largest proportion of parents declared that before the pandemic they spent 2 or fewer hours in front of a computer screen, while during lockdown, this time lasted between 3 and 4 h (62.5% and 34.4%, respectively). Both before and during the pandemic, the majority of parents reported sleeping between 7 and 9 h (53.5% and 58.8%, respectively). Both before the pandemic and during lockdown, most parents admitted to not undertaking any physical activity (47.3% and 43.0%, respectively). Most children spent 30–60 min in front of a computer screen before the pandemic. During lockdown, this time exceeded 120 min (37.2% and 44.2%, respectively). The majority of children slept fewer than 10 h daily before the pandemic. During lockdown, children’s sleep duration ranged between 10 and 13 h (52.9% and 61.8%, respectively). Both before the pandemic and during lockdown, most children were physically active for over 60 min during the day (51.2% and 40.7%, respectively).

Among parents, statistically significant differences were found in the time spent in front of a computer screen and the daily duration of sleep between the periods under investigation. Among children, such differences were observed in the time spent in front of a computer screen, daily sleeping hours and the number of minutes devoted to daily physical activity (Table 4). Among parents, statistically significant differences were found in the time spent in front of a computer screen and the daily duration of sleep between the periods under investigation. Among children, such differences were observed in the time spent in front of a computer screen, daily sleeping hours and the number of minutes devoted to daily physical activity (Table 4).

### 3.5. Correct Parents’ and Children’s Behavior

Figure 1 and Figure 2 present correct parents’ and children’s behaviour.

The most common correct behaviour of parents was occasional consumption of fast food products (76.4% before the pandemic, 79.1% during lockdown) and occasional consumption of sweetened drinks (72.4% and 73.8%, respectively). The least frequent correct behaviour was drinking at least 6 glasses of water daily (24.3% before the pandemic, 30.6% during lockdown) and eating vegetables a few times a day (28.5% and 30.4%, respectively).

Statistically significant differences were found in the frequency of correct behaviour of adults before the pandemic and during lockdown in terms of: eating 4–5 meals daily (*p* = 0.001), drinking the recommended 6 or more glasses of water during the day (*p* = 0.001), eating sweets occasionally or less frequently (*p* = 0.03), consuming no alcohol (*p* = 0.001), sleeping 7–9 h daily (*p* = 0.001) and using electronic devices (computer, television) for <2 h daily (apart from work) (*p* = 0.001) (Figure 1).

The most common correct behaviour of children was occasional consumption of fast food products (86.9% before the pandemic, 86.8% during lockdown) and occasional consumption of sweetened drinks (86.7% and 84.8%, respectively). The least frequent correct behaviour in both periods under investigation was the consumption of vegetables a few times a day (26.1% before the pandemic and 26.8% during lockdown). During the pandemic, the least frequent correct behaviour of children was using electronic devices for less than 1 h daily (21.0%).

Statistically significant differences were found in the frequency of correct behaviour of children before the pandemic and during lockdown in terms of keeping 3–4 h intervals between meals (*p* = 0.01), sleeping 10–13 h daily (*p* = 0.001) and using electronic devices (computer, television) up to 1 h daily (*p* = 0.001) (Figure 2).

### 3.6. The Evaluation of the Lifestyle

The evaluation of the lifestyle of parents and children during lockdown is presented in Figure 3.

Among the parents taking part in the study, during lockdown, 11.2% had a healthy lifestyle, 56.9% moderately healthy lifestyle and 31.8% had an unhealthy lifestyle. Among children, 12.1% had a healthy lifestyle, 68.3% moderately healthy lifestyle and 19.6% an unhealthy lifestyle (Figure 3).

## 4. Discussion

Due to changes in the lifestyle of parents and children, it is necessary to compare the results of our own research with the results of other authors, which will allow for draw conclusions and make recommendations for the future.

The first to consider was the regularity of eating meals. The results of the current study demonstrated that the majority of parents ate 4–5 meals daily both before the pandemic and during lockdown, with the figure rising from 50.5% to 66.0%. Slight changes in the number of meals were observed by Cheikh Ismail et al., who investigated changes in the lifestyle and eating habits of 1012 adult residents of the United Arab Emirates [32]. Before the pandemic, 51.5% of the subjects ate 3–4 meals during the day. During the pandemic, 56.5% of the study participants ate meals with such frequency. As far as children are concerned, in the current study, most of them ate 5 meals daily in both periods (50.1% before and 49.2% during lockdown). Di Renzo et al., who studied lifestyle changes caused by lockdown in Italians aged 12 or more years, did not observe any changes in the number of meals ate by children, similar to the present study [33].

Another issue that was analyzed by authors was the consumption selected healthy group products: vegetables and fruit. Parents usually consumed vegetables a few times a day (28.5% before and 30.2% during lockdown). However, Zhao Hu et al., who studied the impact of the pandemic on the lifestyle of 1033 individuals living in mainland China, found an increase in vegetable consumption among adults [34]. The authors observed a 30% increase in the consumption of vegetables during the pandemic. In the current study, no significant differences were found regarding vegetable consumption among children. Pietrobelli et al., who investigated the influence of the COVID-19 pandemic on obese children living in Verona, also did not observe any significant differences in vegetable consumption by children between the two periods in question (*p* = 0.438) [35]. Fruit consumption among parents did not significantly differ between the two periods and most parents ate fruit once a day (30.6% before and 28.6% during lockdown). Zhao Hu et al. obtained different results in their study [34]. They observed the frequency of fruit consumption to rise during the pandemic among adults. As for children, the largest proportion of them consumed fruit once a day before the pandemic (32.9%) and a few times a day during the pandemic (32.2%). Fruit consumption was significantly different among children in the periods under investigation. Different results in this respect were obtained by López-Bueno et al., who studied health behaviour of Spanish children aged from 3 to 16 years during lockdown [36]. The authors observed a significant decrease in fruit consumption (*p* < 0.001). Low and insufficient fruit consumption makes the diet low in polyphenols, vitamins and other nutrients valuable for health.

There was also analyzed the frequency of consumption of non-recommended products: containing simple sugars, trans fats and alcohol. The frequency of sweet consumption by children in the present study differed between the periods in question; however, most subjects usually ate sweets a few times a week (26.7% before and 26.9% during lockdown). Hashem et al., who studied the impact of lockdown on 765 children living in Egypt, demonstrated that the consumption of sweets and unhealthy products increased among 45.6% of the children during the period in question [37]. The increase of the sweets consumption may have been caused due to the way in which various cultural groups deal with stress. In the current study, before and during the pandemic, most parents consumed fast food products occasionally (60.6% and 45.6%, respectively). Similar results were obtained by Flanagan et al., who studied the impact of lockdown on dietary behaviour of 7753 adults living in the United States, Australia and the United Kingdom [38]. According to the authors, during the pandemic, the consumption of fast food products decreased. The decrease of the fast food consumption may have been caused due to the restriction of availability to restaurants. During the lockdown there was the prohibition of ordering and eating meals inside the restaurant and it was only possible to order take-out food.

Alcohol consumption by adults significantly differed between the periods before and during the pandemic. Before the pandemic, the largest proportion of subjects drank alcohol occasionally (46.2%), while during the pandemic and the associated lockdown, most of the subjects stopped consuming alcohol altogether (42.6%). Grossman E et al., who studied alcohol consumption in 832 women living in the United States, found the opposite: 60% of the subjects indicated that they increased their alcohol intake during the pandemic, while 15% of the study participants stated that their consumption decreased [39]. Increasing the frequency of alcohol consumption contributes to emotional disturbance, which can induce stress intolerance which is especially dangerous during a lockdown when mental health of the population is at risk [21].

The remaining, non-nutritional elements of the family’s lifestyle that authors analyzed were: physical activity, screening time and amount hours of sleep. Parents’ electronic device use time increased during the pandemic. Similar results were obtained by Górnicka et al., who investigated dietary and lifestyle changes in 2575 adult Polish people during lockdown [40]. As for the parents’ duration of sleep, the present study demonstrated an increase in the percentage of individuals who slept for more than 9 h daily (2.2% before the pandemic, 10.7% during lockdown). Among children, the majority slept for less than 10 h before the pandemic (52.9%) and for 10–13 h during lockdown (61.8%). In their international study, AMHSI Research Team, investigated the impact of lockdown on the sleep of 14,000 individuals coming from 11 countries [41]. Similar to our study, the researchers observed sleeping time to increase from 7 h 50 min to 9 h 10 min. Similar results were also obtained by Romdhani et al., Who examined the sleep quality of 3911 elite athletes from 49 countries. Among people practicing competitive sports, the quality of sleep has deteriorated and, what is worrying, the circadian rhythm of athletes has been disturbed [25]. Regarding physical activity, it was found that most parents did not undertake any physical activity in both investigated periods (47.3% before and 43.0% during lockdown). Castañeda-Babarro et al. studied changes in physical activity of 3800 adult Spaniards during the pandemic [42]. They found that the percentage of individuals that were physically active for the recommended 75 min decreased by 10.7% compared to the time before the pandemic. Among children, in both periods, the largest proportion of participants performed physical activity for more than 60 min per day. However, the figure decreased during the pandemic (from 51.2% before to 40.7% during lockdown). Moore et al. studied movement and play behaviour of 1472 Canadian children during the COVID-19 pandemic [43]. Similar to the present study, the researchers observed a significant decrease in physical activity, particularly that undertaken outdoors; at the same time, they found screen time to increase. Both sleep and physical activity during pandemic isolation were the topic of study of Trabelsi et al. In their global analysis, researchers found that during isolation, the quality of sleep worsened and the time spent on physical activity decreased, as opposed to the time spent daily-sitting, which often involves using electronic devices [3]. Extending the time of electronic devices use and reducing the amount of physical activity are behaviors that are particularly dangerous to health. Those bad habits predispose to the development of chronic diseases, contributes to weight gain resulting in overweight, and to orthopedic problems. Incorrect behaviors in this area are the cause of the civilization diseases of the 21st century [28,29].

The results of the present study and those of authors from around the world indicate that the pandemic and the associated lockdowns are having a significant impact on the lifestyle of adults and children, affecting dietary behaviour and other aspects of lifestyle.

## 5. Conclusions

A moderately healthy lifestyle was predominant among families with children aged 4–8 years during lockdown. The parents’ lifestyle during lockdown significantly changed in relation to the time before the pandemic in terms of meal regularity, frequency of vegetable, fast food, sweetened drink and alcohol consumption, electronic device use time and daily sleep duration. As far as children are concerned, their lifestyle changed in terms of meal regularity, frequency of fruit, sweet, savoury snack and fast food consumption, electronic device use time, and daily duration of sleep and physical activity.

This study is important for the development of high-quality public health internationally and as well in the Polish context. It indicates lifestyle dangers that may occur among families with children when it is necessary to introduce restrictions in connection with the COVID-19 pandemic in the future. The study indicates the need for health education in the field of healthy management of excess of free time at home resulting from the isolation. It also points the need for nutritional education, which should encourage the use of a varied diet, which also contributes to the improvement of the immunity, very important factor in COVID-19 context.

### 5.1. Strength

The study was conducted in the first period of the blockade caused by the COVID-19 pandemic in Poland. Isolation has not been known before, so its potential impact on lifestyle changes has not yet been studied. It constitutes the strength of the research and allows for the design of educational activities in the field of coping with such situations.

### 5.2. Limitations

Limitations should be carefully considered when interpreting the results of this study. The survey covered parents of children from all over Poland, but the respondents cannot be divided according to their place of residence. The confidence intervals that will be included in future authors’ publications on lifestyle have also not been defined.

### 5.3. Implications

The results of this study indicate the need of monitoring the lifestyle of parents and children after the lockdown periods and after the COVID-19 pandemic. It may be particularly important to assess the lifestyle of children in the educational institutions they attend. Such research will allow the introduction of educational programs for families and prevent the incorrect habits in adulthood.

## Figures and Tables

**Figure 1 ijerph-19-15064-f001:**
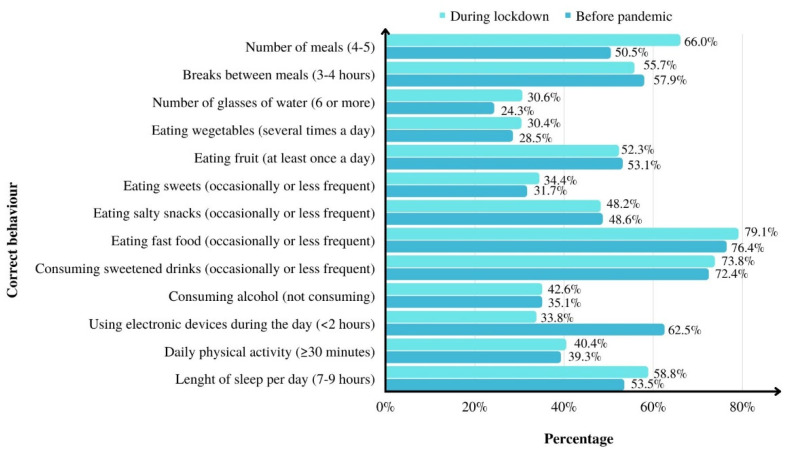
Percentage of parents’ correct behaviour.

**Figure 2 ijerph-19-15064-f002:**
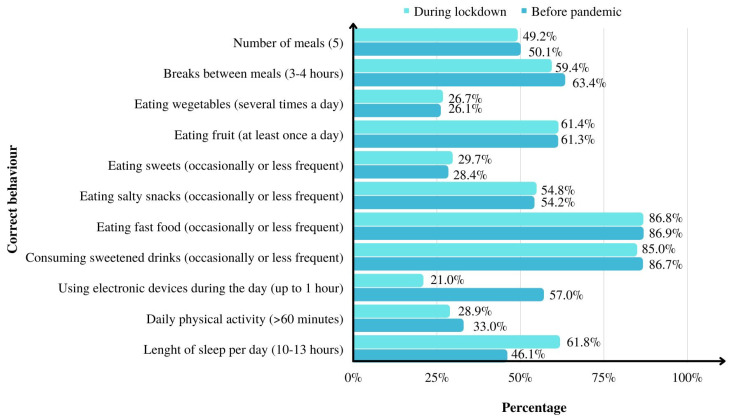
Percentage of children’s correct behaviour.

**Figure 3 ijerph-19-15064-f003:**
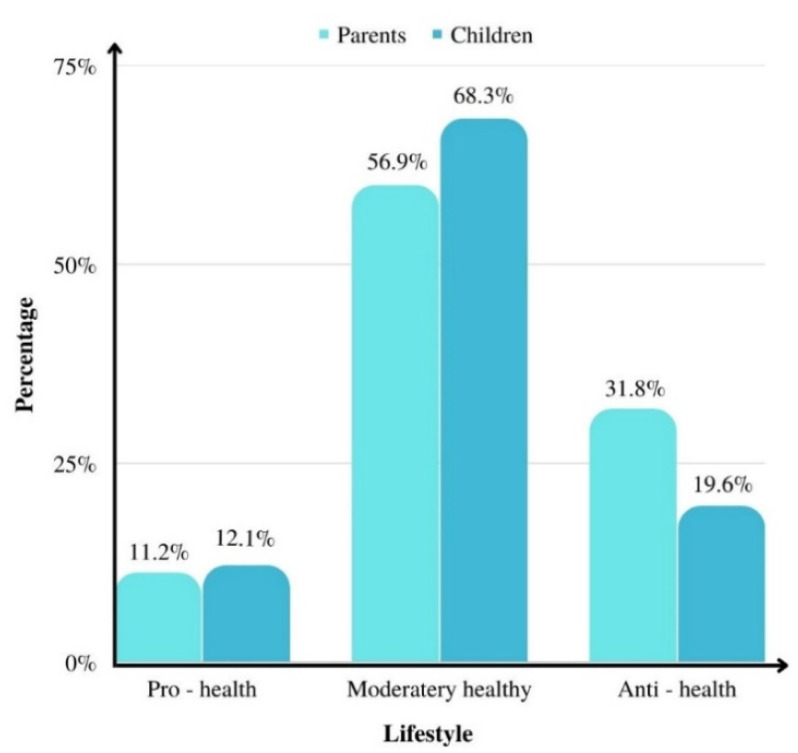
Evaluation of parents’ and children’s lifestyle.

**Table 1 ijerph-19-15064-t001:** Selected meal-related behaviour.

		Before Pandemic		During Locdown		*p*-Value
		N = 1098	%	N = 1098	%	
Parents						
**Number of meals**	Less than 3	132	12.0	40	3.6	0.001 *p* < 0.05
	3	338	35.3	205	18.7	
	4–5	**554**	**50.5**	**725**	**66.0**	
	More than 5	24	2.2	128	11.7	
**Length of breaks between meals**	Less than 3 h	337	30.7	104	9.5	0.001 *p* < 0.05
	3–4 h	**636**	**57.9**	**612**	**55.7**	
	More than 4 h	125	11.3	382	34.8	
**Children**						
**Number of meals**	3 or less	55	5.0	48	4.4	0.001 *p* < 0.05
	4	355	32.3	311	28.3	
	5	**550**	**50.1**	**540**	**49.2**	
	More than 5	138	12.6	199	18.1	
**Length of breaks between meals**	Less than 3 h	345	31.4	390	35.5	0.02 *p* < 0.05
	3–4 h	**697**	**63.5**	**652**	**59.4**	
	More than 4 h	56	5.1	56	5.1	

**Table 2 ijerph-19-15064-t002:** Frequency of consumption of selected products by parents.

The Frequency of Consumption		Before Pandemic		During Locdown		*p*-Value
		N = 1098	%	N = 1098	%	
**Vegetables**	Never	9	0.8	10	0.9	0.04 *p* < 0.05
	Occasionally	223	20.3	197	17.9	
	Several times a month	69	6.3	90	8.2	
	Several times a week	214	19.5	206	18.8	
	Once a day	270	24.6	261	23.8	
	Several times a day	**313**	**28.5**	**334**	**30.2**	
**Fruit**	Never	16	1.5	22	2.0	0.85 *p* > 0.05
	Occasionally	215	19.6	195	17.8	
	Several times a month	83	7.6	102	9.3	
	Several times a week	201	18.3	205	18.7	
	Once a day	**336**	**30.6**	**314**	**28.6**	
	Several times a day	247	22.5	260	23.7	
**Sweets** **(cakes, cookies, candies)**	Never	59	5.4	84	7.7	0.27 *p* > 0.05
	Occasionally	**289**	**26.3**	**294**	**26.8**	
	Several times a month	165	15.0	144	13.1	
	Several times a week	278	25.3	272	24.8	
	Once a day	212	19.3	188	17.1	
	Several times a day	95	8.7	116	10.7	
**Salty snacks** **(chips, peanuts, sticks)**	Never	122	11.1	166	15.1	0.90 *p* > 0.05
	Occasionally	**412**	**37.5**	**363**	**33.1**	
	Several times a month	244	22.2	232	21.1	
	Several times a week	210	19.1	231	21.0	
	Once a day	82	7.5	70	6.4	
	Several times a day	28	2.6	36	3.3	
**Fast food products (kebab, pizza, hamburger)**	Never	173	15.8	367	33.4	0.001 *p* < 0.05
	Occasionally	**666**	**60.7**	**501**	**45.6**	
	Several times a month	221	20.1	171	15.6	
	Several times a week	30	2.7	49	4.5	
	Once a day	6	0.6	5	0.5	
	Several times a day	2	0.2	5	0.5	
**Alcohol** **(beer, wine, vodka)**	Never	385	35.1	**468**	**42.6**	0.44 *p* < 0.05
	Occasionally	**507**	**46.2**	411	37.4	
	Several times a month	145	13.2	122	11.1	
	Several times a week	50	4.6	78	7.1	
	Once a day	9	0.8	13	1.2	
	Several times a day	2	0.2	6	0.6	

**Table 3 ijerph-19-15064-t003:** Frequency of consumption of selected products by children.

The Frequency of Consumption		Before Pandemic		During Lockdown		*p*-Value
		N = 1098	%	N = 1098	%	
**Vegetables**	**Never**	42	3.8	44	4.0	0.06 *p* > 0.05
	Occasionally	191	17.4	175	15.9	
	Several times a month	82	7.5	71	6.5	
	Several times a week	198	18.0	215	19.6	
	Once a day	**298**	**27.1**	**299**	**27.2**	
	Several times a day	287	26.1	294	26.8	
**Fruit**	Never	17	1.6	21	1.9	0.001 *p* < 0.05
	Occasionally	180	16.4	157	14.3	
	Several times a month	77	7.0	82	7.5	
	Several times a week	151	13.8	164	14.9	
	Once a day	**361**	**32.9**	320	29.1	
	Several times a day	312	28.4	**540**	**32.2**	
**Sweets** **(cakes, cookies, candies)**	Never	33	3.0	40	3.6	0.02 *p* < 0.05
	Occasionally	279	25.4	286	26.1	
	Several times a month	193	17.9	156	14.2	
	Several times a week	**293**	**26.7**	**295**	**26.9**	
	Once a day	241	22,0	228	20.8	
	Several times a day	59	5.4	93	8.5	
**Salty snacks** **(chips, peanuts, sticks)**	Never	176	16.0	192	17.5	0.001 *p* < 0.05
	Occasionally	**419**	**38.2**	**410**	**37.3**	
	Several times a month	242	22.0	203	18.5	
	Several times a week	177	16.1	193	17.6	
	Once a day	66	6.0	75	6.8	
	Several times a day	18	1.6	25	2.3	
**Fast food products (kebab, pizza, hamburger)**	Never	424	38.6	**500**	**45.5**	0.001 *p* < 0.05
	Occasionally	**530**	**48.3**	453	41.3	
	Several times a month	122	11.1	110	10.0	
	Several times a week	18	1.6	27	2.5	
	Once a day	2	0.2	5	0.5	
	Several times a day	2	0.2	3	0.3	
**Sweetened drinks (Coca-Cola, flavored waters)**	Never	**605**	**55.1**	**614**	**55.9**	0.04 *p* < 0.05
	Occasionally	347	31.6	319	29.1	
	Several times a month	85	7.7	86	7.8	
	Several times a week	32	2.9	37	3.4	
	Once a day	14	1.3	23	2.1	
	Several times a day	15	1.4	19	1.7	

**Table 4 ijerph-19-15064-t004:** Selected aspects of parents’ and children’s lifestyles.

Selected Element/Day		Before Pandemic		During Lockdown		*p*-Value
		N = 1098	%	N = 1098	%	
Parents						
**Number of hours spent in front of the computer, TV (excluding work)**	2 or less	**686**	**62.5**	371	33.8	0.001 *p* < 0.05
	3-4	298	27.1	**378**	**34.4**	
	>4	114	10.4	349	31.8	
**Hours of sleep**	≤5	103	9.4	76	6.9	0.001 *p* < 0.05
	6	384	35.0	259	23.6	
	7–9	**587**	**53.5**	**646**	**58.8**	
	>9	24	2.2	117	10.7	
**Number of minutes spent on physical activity**	not undertaking	**519**	**47.3**	**472**	**43.0**	0.16 *p* > 0.05
	<15	148	13.5	182	16.6	
	15–30	177	16.1	183	16.7	
	>30–60	171	15.6	191	17.4	
	>60	83	7.6	70	6.4	
**Children**						
**Number of minutes spent in front of the computer, TV**	<30	218	19.9	66	6.0	0.001 *p* < 0.05
	30–60	**408**	**37.1**	164	14.9	
	>60–120	349	31.8	383	34.9	
	>120	123	11.2	**485**	**44.2**	
**Hours of sleep**	<10	**581**	**52.9**	399	36.3	0.001 *p* < 0.05
	10-13	506	46.1	**679**	**61.8**	
	>13	11	1.0	20	1.8	
**Number of minutes spent on physical activity**	not undertaking	44	4.0	95	8.7	0.001 *p* < 0.05
	<30	130	11.8	228	20.8	
	30–60	362	33.0	328	29.9	
	>60	**562**	**51.2**	**447**	**40.7**	

## Data Availability

The data presented in this study are available on request from the corresponding author.

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
