# Peer review of "Lifestyle of Families with Children Aged 4–8 Years before and during Lockdown Due to COVID-19 Pandemic in Poland"

_ijerph, 2022, doi:10.3390/ijerph192215064_

Round 1

Reviewer 1 Report

Dear Authors!

The manuscript is timely, well written, and may further benefit from some modifications. Therefore, I recommend revisions based on my feedback/comments below. Thank you!

ABSTRACT

Line 12: I think the following is not optimal, please revise: “An originally developed questionnaire was used as the research tool” (please see lines 99-100 as well).

INTRODUCTION

Line 23-29: Suggest to incorporate this article:

https://pubmed.ncbi.nlm.nih.gov/34937958/

Line 72-78: Suggest to discuss/incorporate this article here:

https://pubmed.ncbi.nlm.nih.gov/34878639/

Line 96: Do you have the study’s hypothesis?

Line 103-104: The sentence appears sub-optimal, please revise.

Line 116-120: Which guideline was followed for the thresholds used (40% etc).?

MATERIALS AND METHODS

It is not very clear how exactly the data were collected?

Do you have sample size calculation and justification?

What steps were taken to clean the data? E.g., data should be screened for attention patterns, click-through behaviours, keystroke analysis, machine responses etc.

The language of data collection was in Polish or multiple languages? Suggest to add the procedures of translation, re-translation etc. and the protocol was followed?

RESULTS

Line 166: The following is overlapping (i.e., 30 mins): “Number of minutes spent on physical activity” = 15-30 vs 30-60? How to decide if the duration is 30 mins? Same for the next question (60 vs 60 mins). Please, check the whole manuscript.

Strictly speaking, I’d add confidence interval, i.e., % and CI presentation of results. Why not being considered?

Only one decimal is needed, e.g., 80.0% instead of 80.04% (i.e., minimum adequate accuracy). Please consider this throughout the manuscript. Just like what you have done for Figure 1 (but not in Figure 2 etc.)

https://onlinelibrary.wiley.com/doi/10.1111/j.1600-0838.2011.01393.x

I struggle to distinguish the following: “Occasionally vs Several times a month”. Can you sort it our please?

Line 198 and so on: The p=0.000 should be changed to p<0.001 (in text and tables, throughout the manuscript)

DISCUSSION

Line 227: The first paragraph of the Discussion section is not for this explanation. Please, replace this paragraph with your major results and a brief concluding remark.

In Discussion, when you start a new paragraph, please set the topic first, e.g., usually the main findings. I.e., use the key findings outlined in the first paragraph of Discussion to start the subsequence (new) paragraphs in the Discussion. You then start to discuss accordingly and critically. For clarity, can you please consider this?

Do you have “strengths and limitations”?

Can you check the reference please; appears inconsistent and different from MDPI guideline.

Thank you.

Author Response

Dear Reviewer 

We would like to kindly thank you for the review. We appreciate your comments and we would like to response.  

In response to your review: 

ABSTRACT: “Line 12: I think the following is not optimal, please revise: “An originally developed questionnaire was used as the research tool” (please see lines 99-100 as well).”
- This has been corrected as recommended. 

INTRODUCTION:  

“Line 23-29: Suggest to incorporate this article: https://pubmed.ncbi.nlm.nih.gov/34937958/; Line 72-78: Suggest to discuss/incorporate this article here: https://pubmed.ncbi.nlm.nih.gov/34878639/
- These articles have been incorporated as recommended.

“Line 96: Do you have the study’s hypothesis?”
- We added the research hypothesis. It was assumed that the families' lifestyles deteriorated during the quarantine.

“Line 103-104: The sentence appears sub-optimal, please revise.”
- The sentence was corrected.

“Line 116-120: Which guideline was followed for the thresholds used (40% etc).?”
- The
"material and methods" part was expanded and the following sentence was included: The families' lifestyle was assessed on the basis of the above guidelines, where each correct eating behavior was awarded 1 point and the incorrect eating behavior was not scored. 

MATERIALS AND METHODS 

“It is not very clear how exactly the data were collected? Do you have sample size calculation and justification? What steps were taken to clean the data? E.g., data should be screened for attention patterns, click-through behaviours, keystroke analysis, machine responses etc. The language of data collection was in Polish or multiple languages? Suggest to add the procedures of translation, re-translation etc. and the protocol was followed?”
- The entire "material and methods" section has been extensively developed and detailed in line with the recommendations.

RESULTS 

“Line 166: The following is overlapping (i.e., 30 mins): “Number of minutes spent on physical activity” = 15-30 vs 30-60? How to decide if the duration is 30 mins? Same for the next question (60 vs 60 mins). Please, check the whole manuscript.”
-
Thank you. We made corrections: possible answers regarding physical activity in the questionnaire were: not undertaking, <15, 15-30,> 30-60,> 60 and regarding number of minutes spent in front of the computer, TV: <30, 30-60, > 60-120,> 120. 

Strictly speaking, I’d add confidence interval, i.e., % and CI presentation of results. Why not being considered?
- The authors considered this as a limitation of the study.

“Only one decimal is needed, e.g., 80.0% instead of 80.04% (i.e., minimum adequate accuracy). Please consider this throughout the manuscript. Just like what you have done for Figure 1 (but not in Figure 2 etc.) https://onlinelibrary.wiley.com/doi/10.1111/j.1600-0838.2011.01393.x”; “Line 198 and so on: The p=0.000 should be changed to p<0.001 (in text and tables, throughout the manuscript)”
- It has been corrected as recommended.

I struggle to distinguish the following: “Occasionally vs Several times a month”. Can you sort it our please?
- In the authors' concept and in other polish study, the "occasionally" answer means less frequent than several times a month, for example 2 times a year, at the family party. 

DISCUSSION 

“Line 227: The first paragraph of the Discussion section is not for this explanation. Please, replace this paragraph with your major results and a brief concluding remark. In Discussion, when you start a new paragraph, please set the topic first, e.g., usually the main findings. I.e., use the key findings outlined in the first paragraph of Discussion to start the subsequence (new) paragraphs in the Discussion. You then start to discuss accordingly and critically. For clarity, can you please consider this?”
- The "discussion" section has been corrected as recommended.

“Do you have “strengths and limitations”?”
-We added the strengths, limitations and implications of the study:

5.1 Strength  The study was conducted in the first period of the blockade caused by the COVID-19 pandemic in Poland. Isolation has not been known before, so its potential impact on lifestyle changes has not yet been studied. It constitutes the strength of the research and allows for the design of educational activities in the field of coping with such situations.    

5.2 Limitations should be carefully considered when interpreting the results of this study. The survey covered parents of children from all over Poland, but the respondents cannot be divided according to their place of residence. The confidence intervals that will be included in future authors' publications on lifestyle have also not been defined.    

5.3 Implications  The results of this study indicate the need of monitoring the lifestyle of parents and children after the lockdown periods and after the COVID-19 pandemic. It may be particularly important to assess the lifestyle of children in the educational institutions they attend. Such research will allow the introduction of educational programs for families and prevent the incorrect habits in adulthood. 

 “Can you check the reference please; appears inconsistent and different from MDPI guideline.”
- References have been revised as recommended.

We would like to thank you again for the review.  

Yours faithfully  
The authors 

Reviewer 2 Report

Thank you for submitting this manuscript. There are some problems with some parts of this manuscript. Please note these items:

Method:

The research method should be edited. Data collection was done with a researcher-made questionnaire. But it is not clear how the research tool was prepared and based on what reference. Is the research tool design approach deductive or inductive? How to determine its validity and reliability is not mentioned. How was the distribution of research tools? When was the sampling done? Is informed consent obtained? How were the families identified? How is the sample size determined?

Findings:

In the findings, many tables and figures are drawn, while the focus should be on the most important findings of the study. The findings should be brief and expressed based on the research objectives.

Discuss:

The discussion should be written more precisely. According to the results of other similar studies, a comparison should be made and a logical conclusion should be mentioned.

Conclusion:

 What were the limitations of this research? What are some suggestions for further studies?

Author Response

Dear Reviewer 

We would like to kindly thank you for the review. We appreciate your comments and we would like to response.  

In response to your review:

  1. Method: “The research method should be edited. Data collection was done with a researcher-made questionnaire. But it is not clear how the research tool was prepared and based on what reference. Is the research tool design approach deductive or inductive? How to determine its validity and reliability is not mentioned. How was the distribution of research tools? When was the sampling done? Is informed consent obtained? How were the families identified? How is the sample size determined?”  

- In line with the comments, extensive changes have been made to the "Materials and methods" section. This section has been extensively developed and detailed in line with the recommendations. 

  1. Findings: “In the findings, many tables and figures are drawn, while the focus should be on the most important findings of the study. The findings should be brief and expressed based on the research objectives.”  

- The authors planned an extensive study on the lifestyle of families. Among the obtained results, the ones that characterize the lifestyle most accurately (nutritional behavior and non-nutritional elements) of both parents and children are presented. For this reason, there are more tables in the publication than in the standard ones.   

  1. Discussion: “The discussion should be written more precisely. According to the results of other similar studies, a comparison should be made and a logical conclusion should be mentioned.” 

- The "discussion" section has been corrected in line with the comments. An introduction to individual paragraphs was introduced and its content expanded. 

  1. Conclusion: “ What were the limitations of this research? What are some suggestions for further studies?”  

- Thank you. We added the strengths, limitations and implications of the study:  

5.1 Strength: The study was conducted in the first period of the blockade caused by the COVID-19 pandemic in Poland. Isolation has not been known before, so its potential impact on lifestyle changes has not yet been studied. It constitutes the strength of the research and allows for the design of educational activities in the field of coping with such situations.    

5.2  Limitations should be carefully considered when interpreting the results of this study. The survey covered parents of children from all over Poland, but the respondents cannot be divided according to their place of residence. The confidence intervals that will be included in future authors' publications on lifestyle have also not been defined.    

5.3 Implications: The results of this study indicate the need of monitoring the lifestyle of parents and children after the lockdown periods and after the COVID-19 pandemic. It may be particularly important to assess the lifestyle of children in the educational institutions they attend. Such research will allow the introduction of educational programs for families and prevent the incorrect habits in adulthood. 

We would like to thank you again for the review.  
Yours faithfully  
The Authors 

Reviewer 3 Report

Thank you for the opportunity to review this interesting research results. The topic is of high relevance and importance.

I take this opportunity to make some comments to authors. Please see the attachment.

Author Response

We would like to kindly thank you for the review. 

Yours faithfully  
The authors 

Round 2

Reviewer 1 Report

The manuscript has been revised, largely according to the suggestions and comments.

Only one note.

For % presentation, only one decimal is needed, e.g., 60.0% instead of 60.03% , so please consider "minimum adequate accuracy", and update throughout.

Thank you.

Author Response

Dear Reviewer 

We would like to kindly thank you. We have made corrections. 

Yours faithfully  
The authors 

Reviewer 2 Report

I appreciate that the authors took great care to respond point by point to the reviewer's comments. I think the article is ready for publication.

Author Response

Dear Reviewer 

We would like to kindly thank you. 

Yours faithfully  
The authors